# Imidazole-Based pH-Sensitive Convertible Liposomes for Anticancer Drug Delivery

**DOI:** 10.3390/ph15030306

**Published:** 2022-03-03

**Authors:** Ruiqi Huang, Vijay Gyanani, Shen Zhao, Yifan Lu, Xin Guo

**Affiliations:** Thomas J. Long School of Pharmacy, University of the Pacific, Stockton, CA 95211, USA; r_huang9@u.pacific.edu (R.H.); v_gyanani@u.pacific.edu (V.G.); s_zhao5@u.pacific.edu (S.Z.); y_lu5@u.pacific.edu (Y.L.)

**Keywords:** pH-sensitive, liposome, imidazole, anticancer, drug delivery, multicellular spheroids

## Abstract

In efforts to enhance the activity of liposomal drugs against solid tumors, three novel lipids that carry imidazole-based headgroups of incremental basicity were prepared and incorporated into the membrane of PEGylated liposomes containing doxorubicin (DOX) to render pH-sensitive convertible liposomes (ICL). The imidazole lipids were designed to protonate and cluster with negatively charged phosphatidylethanolamine-polyethylene glycol when pH drops from 7.4 to 6.0, thereby triggering ICL in acidic tumor interstitium. Upon the drop of pH, ICL gained more positive surface charges, displayed lipid phase separation in TEM and DSC, and aggregated with cell membrane-mimetic model liposomes. The drop of pH also enhanced DOX release from ICL consisting of one of the imidazole lipids, *sn*-2-((2,3-dihexadecyloxypropyl)thio)-5-methyl-1H-imidazole. ICL demonstrated superior activities against monolayer cells and several 3D MCS than the analogous PEGylated, pH-insensitive liposomes containing DOX, which serves as a control and clinical benchmark. The presence of cholesterol in ICL enhanced their colloidal stability but diminished their pH-sensitivity. ICL with the most basic imidazole lipid showed the highest activity in monolayer Hela cells; ICL with the imidazole lipid of medium basicity showed the highest anticancer activity in 3D MCS. ICL that balances the needs of tissue penetration, cell-binding, and drug release would yield optimal activity against solid tumors.

## 1. Introduction

The past two decades have seen a surge in the medical use of nanotechnology. Tens of nanomedicines, mostly liposomes, have been approved by Food and Drug Administration (FDA) or European Medicines Agency (EMA) to treat or diagnose many serious diseases, especially various types of cancer [1,2,3,4]. Liposomal formulations of doxorubicin (DOX) [4,5,6] represent an important group of nanomedicines, which are indicated for a wide range of cancers. Many studies have demonstrated that, compared with free DOX, DOX-loaded nanomedicines offer substantially lower cardiotoxicity and higher efficacy, both owing to their preferred accumulation at tumor sites [7,8].

One abnormal feature in many solid tumors is a fenestrated vasculature [9], which allows nano-formulations of anticancer drugs to permeate selectively from the blood circulation to the tumor interstitium. The nano-formulations can then accumulate in solid tumors due to their lack of lymphatic drainage, a phenomenon known as the enhanced permeability and retention (EPR) effect. Many long-circulating nano-formulations have been developed to take advantage of the EPR effect, including PEGylated liposomes, hydrophilic polymers, and solid lipid nanoparticles [10,11]. However, the fenestrated vasculature distributes mainly in the peripheral of solid tumors, which could limit the distribution of nano-drug formulations to the tumor core, and hence limit their ability to eradicate the entire tumor cell population [12].

Another abnormality of tumor tissue is its acidic microenvironment. Whereas normal cells in healthy tissues have an intracellular pH (pH_i_) of 7.2 and a slightly higher extracellular pH (pH_e_) of 7.4, cancer cells in tumors are characterized by a pH_i_ of 7.2 but a significantly lower pH_e_ of 6.2–7.0 [13,14]. The lower pH_e_ in the tumor interstitium results from the accumulation of lactate, an acidic by-product of the elevated anaerobic metabolism by the cancer cells in the hypoxic tumor microenvironment [15]. In response, many pH-sensitive drug delivery systems have been developed, including pH-sensitive liposomes, antibody-drug conjugates with acid-labile linkers, and pH-sensitive polymeric nanoparticles [16,17,18]. A number of pH-sensitive drug delivery systems have shown enhanced anticancer activity compared to their pH-insensitive counterparts in preclinical research [19,20]. However, pH-sensitive nano-drug delivery systems have not yet been approved to treat cancer patients.

Imidazole represents an important pH-sensitive functional group in pharmaceutical sciences. Imidazoles carry pKa values around 5.0–6.5 and thus can protonate to assume a positive charge in response to weakly acidic pH in pathophysiological settings [21]. The incorporation of imidazole-based lipids into nano-formulations has enhanced their intracellular delivery of proteins and nucleic acids [22,23]. However, very few studies [24] have been reported on imidazole-based nano-formulations for anticancer drug delivery.

Herein, we report a novel type of imidazole lipids and their pH-sensitive liposomes. Our goal is to develop imidazole lipids that trigger the liposomes in cooperation with phosphatidylethanolamine-polyethylene glycol conjugates (PE-PEG), which is a key component to stabilize liposomes in blood circulation for anticancer drug delivery. At pH 7.4, the imidazole lipids are mostly uncharged, while at acidic pH, they would protonate and cluster with negatively charged PE-PEG to induce lipid phase conversions. Such liposomes are thus called imidazole-based convertible liposomes (ICL) (Figure 1). ICL was loaded with the anticancer drug doxorubicin (DOX) and subjected to physicochemical and morphological characterizations. The pH-sensitivity of ICL was assessed by Differential Scanning Calorimetry (DSC), change of ζ-potential, interaction with negatively charged model lipid membranes, and pH-dependent drug release. The anticancer activities of ICL were assessed against both 2D monolayer cancer cells and 3D multicellular tumor spheroids (MCS), which mimic more features of solid tumors than 2D cell cultures, including the acidic microenvironment, the dense ECM, and the hypoxic core [9,25]. PEGylated liposomes containing DOX but not the pH-sensitive, imidazole-based lipids were also studied, both as a pH-insensitive control and as a benchmark of clinically used liposomal formulations. The effects of cholesterol on the physicochemical properties, pH-sensitivity and anticancer activities of ICL were also investigated. We report the physicochemical properties and the superior anticancer activities of ICL in both monolayer cancer cells and MCS of multiple cancer cell lines in correlation to their pH-sensitivity.

## 2. Results

### 2.1. Imidazole-Based pH-Sensitive Lipids

Three novel lipids (Figure 2), namely *sn*-2-((2,3-dihexadecyloxypropyl)thio)-1H-imidazole (DHI), *sn*-2-((2,3-dihexadecyloxypropyl)thio)-5-methyl-1H-imidazole (DHMI), and *sn*-2-((2,3-dihexadecyloxypropyl)thio)-4,5-dimethyl-1H-imidazole (DHDMI) were designed as a critical component of ICL. Each of the lipids contains two saturated hexadecyl (C16) hydrocarbon chains as the tail and an imidazole-based headgroup. In a typical anticancer liposome formulation, the C16 chains would make the novel lipids more compatible with other lipids, which would also carry long, saturated hydrocarbon chains to enhance the physicochemical stability of the formulation. All the imidazole-based headgroups of the three lipids are expected to interact with the negatively charged DPPE-PEG at lowered pH, but they would each have slightly different pH-sensitivity due to their different substituents. Specifically, the imidazole headgroup of DHI, DHMI, and DHDMI carries zero, one, and two electron-donating methyl groups, respectively, which would yield incrementally higher pKa of the lipids (Figure 2). ICL containing such lipids would therefore be triggered at incrementally higher pH. The three lipids were all synthesized from dihexadecyl glycerol (DHG) in two steps: first, its activation into DHG-tosylate, and then conjugation with the appropriately methylated mercaptoimidazole (Figure 3).

### 2.2. Composition of Imidazole-Based Convertible Liposomes (ICL)

The imidazole-based lipids (DHI, DHMI, and DHDMI) were each mixed with 1,2-distearoyl-*sn*-glycero-3-phosphocholine (DSPC) and 1,2-dipalmitoyl-*sn*-glycero-3-phosphoethanolamine-*N*-[azido(polyethylene glycol)-2000 (DPPE-PEG (2000)) at 25/70/5 molar ratio to construct their corresponding ICL. DSPC is a phospholipid with a neutrally charged choline headgroup and a tail of two long (C18), saturated hydrocarbon chains; DPPE-PEG is a conjugate of PEG2000 and phosphoethanolamine with a negative charge from the phosphate, and a tail of two long (C16) saturated hydrocarbon chains. At pH 7.4, the lipids would assemble into a stable lipid membrane of ICL that is evenly coated by PEG as in sterically hindered, long-circulating liposomes [6,8]. In response to lowered pH, the imidazole-based lipids in ICL would be protonated to assume positive charges and cluster with negatively charged DPPE-PEG (2000) by electrostatic interactions (Figure 1B). Such pH-triggered clustering of lipids would expose part of the ICL surface that is no longer sterically hindered by PEG, which would then enhance the interaction between ICL and cancer cells in the acidic tumor interstitium [26]. The ICL-cancer cell interactions could also be enhanced by the excess positive charges on the ICL surface [26]. Furthermore, the pH-triggered lipid clustering could enhance the release of the liposome contents through the edges between the DPPE-PEG-rich and DPPE-PEG-poor regions of the liposome membrane based on our prior studies on liposomes consisting of pH-sensitive conformational switches of lipid tails [27,28]. As a common lipid component to improve the stability of liposomes, cholesterol [29,30] was included in some of the formulations under investigation. Cholesterol was incorporated at 25 mol%, a level that tends to improve the liposome stability both on the shelf and in blood circulation. Analogous liposomes without imidazole-based lipids were also prepared and characterized as pH-insensitive controls.

### 2.3. Physicochemical Characteristics of ICL

After lipidic film hydration, freeze-anneal-thawing and sequential extrusion through 400 nm, 200 nm and 100 nm polycarbonate membranes, ICL consisting of 25 mol% DHI, DHMI or DHMI were successfully prepared with mean hydrodynamic diameters smaller than 130 nm (Table 1). The Polydispersity Index (PDI), a measure of the heterogeneity of the size of particles in a mixture, was lower than 0.3 for all the liposome formulations. The liposomes with 25 mol% cholesterol showed generally smaller PDI than the cholesterol-free formulations. After being loaded with DOX, all the formulations were characterized with larger sizes and PDI. Nevertheless, the sizes of all the DOX-loaded formulations were below or around 200 nm in diameter. The DOX-loaded ICL with 25% cholesterol showed smaller sizes and PDI than the DOX-loaded ICL without cholesterol. The encapsulation efficiency (EE) of all the formulations was 50% or higher, and ICL with cholesterol were characterized with considerably higher EE than cholesterol-free ICL. 

### 2.4. pH-Triggered Acquisition of Positive Charges on ICL Surface

The *ζ*-potentials of ICL and pH-insensitive liposomes at pH 6.0, 6.5, 7.0, and 7.4, 37 °C were measured to assess the surface charge of ICL in relationship to pH. As shown in Figure 4A, all three ICL consisting of DHI, DHMI, or DHDMI but no cholesterol showed a significant increase of *ζ*-potential when pH was lowered from 7.4 to 6.0. Particularly, the *ζ*-potential of ICL containing DHMI or DHDMI was elevated from below to above zero, indicating the conversion of such ICL to assume positive surface charges in response to the pH drop. Furthermore, the higher the pK_a_ of the imidazole-based lipid (DHDMI > DHMI > DHI), the larger was the increase of the *ζ-*potential of its ICL. This result indicated that the pH-triggered acquisition of positive charges on the ICL surface was rendered by the protonation of the imidazole-based lipids DHI, DHMI and DHDMI. By contrast, the pH-insensitive liposomes (DSPC/DPPE-PEG) displayed negative *ζ-*potentials below −10 mV at both physiological and acidic pH. However, as can be seen in Figure 4B, ICL containing 25 mol% cholesterol didn’t show incremental elevation of *ζ-*potentials at pH 6.0–7.4 but fluctuated between −5 mV and −20 mV, indicating that the pH-sensitivity of ICL, as shown by the pH-triggered acquisition of positive surface charges, was prohibited by the addition of 25 mol% cholesterol.

### 2.5. pH-Triggered Interaction between ICL and Model Liposome

To test if lowered pH can enhance ICL’s interaction with cell membrane, model liposomes (negatively charged, ~200 nm in diameter) that mimicked the composition of the plasma membrane [31] were prepared and mixed with ICL at a 1:1 lipid molar ratio. The mixture was then exposed to pH 7.4, 7.0, 6.5, and 6.0 and characterized by dynamic light scattering (Figure 5). ICL consisting of DHI, DHMI, or DHDMI but no cholesterol (Figure 5A) showed a remarkable increase of diameter from ~200 nm up to ~1300 nm as the pH decreased from 7.4 to 6.0, indicating the aggregation of ICL with the model liposomes in response to the drop of pH. It is worth noting that ICL aggregated with the cell membrane-mimetic liposomes even at near-zero ζ-potentials (DHMI and DHDMI at pH 6.5), indicating that acquisition of excessive positive charge is not necessary for ICL to adsorb onto the model liposomes. Instead, the aggregation may be attributed to the loss of negative charges on the ICL surface when the positively charged imidazole lipids cluster with negatively charged DPPE-PEG. As shown in Figure 5B, the mixture of model liposomes and liposomes consisting of both imidazole-based lipids and cholesterol did not show any size increase, which indicated that the incorporation of 25 mol% cholesterol hindered the interaction between ICL and the model liposomes at the acidic pHs under this investigation.

### 2.6. pH-Triggered Drug Release from ICL

Drug release from ICL and the pH-insensitive control liposomes were monitored at various pHs over 12 h to further characterize the stability and pH-sensitivity of ICL. As shown in Figure 6A–D, the ICL formulations consisting of DHI, DHMI, DHDMI but no cholesterol and the pH-insensitive control liposomes without cholesterol released 64.53 ± 1.74%, 53.65 ± 2.27%, 20.36 ± 0.83% and 29.56 ± 0.70% of the encapsulated DOX, respectively, after incubation at pH 7.4 for 12 h, which indicated that the stability of DHDMI/DSPC/DPPE-PEG and of the pH-insensitive DSPC/DPPE-PEG liposomes was higher than DHI/DSPC/DPPE-PEG and DHMI/DSPC/DPPE-PEG liposomes at the physiological pH 7.4. More importantly, the DHMI/DSPC/DPPE-PEG showed remarkably higher drug release at pH 6.0 than pH 7.4 (~50% vs. ~25%), but the DHI/DSPC/DPPE-PEG, DHMI/DSPC/DPPE-PEG and the pH-insensitive liposomes showed no noticeable enhancement of drug release under the same condition. In the presence of 25 mol% cholesterol, the ICL consisting of DHI, DHMI, DHDMI and the pH-insensitive liposomes released 56.19 ± 0.78%, 47.11 ± 0.60%, 40.32 ± 0.99% and 57.70 ± 3.03% DOX, respectively after incubation at pH 7.4 for 12 h. Compared to their cholesterol-free counterparts, the DHI and the DHMI liposomes with cholesterol (Figure 6E,F) released a similar percentage of DOX at pH 7.4, while the DHDMI (Figure 6G) and the pH-insensitive control liposomes (Figure 6H) released a higher percentage of DOX at pH 7.4. None of the formulations with cholesterol showed enhanced drug release at lowered pH, which indicates that the addition of cholesterol prevented any acidic pH-triggered drug release from ICL.

### 2.7. Morphological Studies on ICL under TEM

Based on the aforementioned three pH-sensitivity studies, the ICL formulations consisting of DHI, DHMI, or DHDMI but no cholesterol were further characterized by TEM (Figure 7). Such ICL formulations showed spherical vesicle structures at pH 7.4, thus confirming the formation of liposomes. While DHI/DSPC/DPPE-PEG and DHMI/DSPC/DPPE-PEG formulations showed vesicles of smooth staining, those of DHDMI/DSPC/DPPE-PEG showed bright patches of light staining at pH 7.4. Upon exposure to lower pH 6.0, all the three ICL formulations showed more bright patches, especially the DHDMI/DSPC/DPPE-PEG formulation, which showed vesicles predominantly with clearly distinguishable bright and dark areas, which most probably represent separated lipid phases that are differentially stained by uranyl acetate. DHMI/DSPC/DPPE-PEG and DHDMI/DSPC/DPPE-PEG ICL also showed larger and brighter vesicles at the lower pH 6.4. Because the positively charged uranyl ions (UO^2+^_2_) of the TEM stain preferably bind to the phosphate groups in lipid bilayers [32], the bright patches probably represent areas of ICL surface where the protonated imidazole-based lipids cluster with phosphate groups of DPPE-PEG and thus hinder their binding with the uranyl ions. At pH 6.0, the DHMI/DSPC/DPPE-PEG formulation also showed some collapsed, non-vesicle structures, which would explain its acidic pH-enhanced release of DOX (Figure 6B).

The mixture of DHMI/DSPC/DPPE-PEG and model liposomes also showed different morphology at pH 7.4 and 6.0. At pH 7.4, the mixture consisted mainly of smoothly stained vesicles; at pH 6.0, the mixture showed both vesicles with bright patches and substantially larger and brighter structures that appear to be aggregates of multiple ICL and model liposomes.

### 2.8. Differential Scanning Calorimetry of ICL

In order to further elucidate the phase behavior of ICL, DHI/DSPC/DPPE-PEG liposomes were characterized by differential scanning calorimetry. As the temperature was gradually increased from 40 °C to 75 °C at pH 7.4, the DSC thermogram of DHI/DSPC/DPPE-PEG liposomes (Figure 8A) showed one broad and tilted peak between 56 °C and 65 °C, which indicated that the liposomes went through mainly one phase transition and therefore started with mainly one lipid phase of a mixture of DHI, DSPC, and DPPE-PEG. At pH 6.0, the DSC thermogram showed two distinct peaks, one at a similar range between 60 °C and 64 °C, and another new peak at around 52 °C (Figure 8A), which indicated the formation of at least two lipid phases in the liposome membrane. The new peak most probably represents a lipid phase that is rich in DSPC because liposomes of only DSPC have a very similar gel-to-liquid phase transition temperature at 54 °C [33]. The DSC thermogram of the pH-insensitive control liposome DSPC/DPPE-PEG (Figure 8B) showed one phase-transition peak around 53 °C at pH 7.4 and 6.0, indicating that the control liposomes had homogenous mixing of lipids at both pHs.

As DSPC carries the longest hydrophobic tail (two C18 chains) among all the lipids of the DHI/DSPC/DPPE-PEG formulation, the phase transition peak in the high-temperature region of 56 °C to 65 °C indicates strong interactions between different types of lipid molecules, most probably DHI and DPPE-PEG, rather than interactions between the same type of lipid molecules in the liposome membrane.

### 2.9. Anticancer Activity of ICL on 2D Monolayer Cells

The anticancer activity of ICL formulations was tested by the decrease of the viability of 2D monolayer cancer cells (HeLa) at both neutral and mildly acidic pH 6.0–6.5 as seen in tumor interstitium. As the culture media pH was adjusted from 7.4 to 6.0, the ICL formulations that consisted of an imidazole-based pH-sensitive lipid but no cholesterol showed higher anticancer activity (Figure 9), especially at 10 μg/mL DOX concentration, where all the ICL formulations under study showed significantly higher anticancer activity (see * in Figure 9) at pH 6.0 than pH 7.4 (*p* < 0.05). Among the three ICL formulations, the DHDMI/DSPC/DPPE-PEG liposomes that contained the imidazole lipid (DHDMI) of the highest pKa showed the best anticancer activity on the monolayer HeLa cells. By contrast, no difference in the anticancer activity of the pH-insensitive control liposomes (DSPC/DPPE-PEG) was detected between pH 7.4 and 6.0. As the positive control, free DOX showed the highest anticancer activity (~50% cell viability at 10 μg/mL), but such activity was the same at both pH 7.4 and pH 6.0.

### 2.10. Anticancer Activity of ICL on 3D Multicellular Spheroids

The ICL formulations were also tested against 3D multicellular spheroids (MCS) of a number of cancer cell lines, including HeLa (cervical cancer), MDA-MB-231 (breast cancer), MDA-MB-468 (breast cancer), and A549 (lung cancer). Compared to monolayer cells, MCS better mimic many features of solid tumors, including a cell cluster structure that hinders drug penetration, a hypoxic microenvironment, and an acidic interstitium. Similar to prior reports [34,35], confocal imaging of MCS in our studies confirmed the presence of pH-gradient from 7–8 at the peripheral to 5.5–6.4 at the core (Appendix A [36]).

After growth to about 500 μm in diameter to ensure the development of a necrotic core and an acid microenvironment, MCS were exposed to incremental concentrations of cholesterol-free ICL, pH-insensitive control liposomes and free DOX for 72 h followed by assessment of the dose-dependent decrease of cancer cell viability. Against HeLa MCS (Figure 10A), the DHI/DSPC/DPPE-PEG and DHMI/DSPC/DPPE-PEG ICL formulations showed significantly better anticancer activity (IC_50_ = 3.82 ± 1.13 and 2.07 ± 1.13 μM, respectively, Table 2) than the pH-insensitive control liposomes (IC_50_ = 11.41 ± 1.28 μM, *p* < 0.001). However, such improvement in anticancer activity was not observed in the DHDMI/DSPC/DPPE-PEG formulation (IC_50_ = 9.51 ± 1.15 μM, *p* = 0.0693, Table 2). Against MDA-MB-468 MCS (Figure 10D), all the three ICL formulations consisting of DHI, DHMI or DHDMI showed better anticancer activity (IC_50_ = 0.38 ± 0.21, 0.31 ± 0.15, and 0.63 ± 0.10 μM, respectively, Table 2) than the pH-insensitive control liposomes (IC_50_ = 1.24 ± 0.13 μM; *p* < 0.001 for DHI and DHMI, *p* < 0.01 for DHDMI) but the improvement was not as large as on HeLa MCS. Against both HeLa and MDA-MB-468 MCS, the DHMI/DSPC/DPPE-PEG ICL showed the best anticancer activity, which was comparable to that of the positive control, free DOX. However, ICL did not show noticeable improvement in activity against A549 (Figure 10B) or MDA-MB-231 (Figure 10C) MCS, compared with pH-insensitive control liposomes. When 25 mol% cholesterol was included in the lipid composition (Figure 10E–G), all the ICL formulations showed similar but not better anticancer activity compared with pH-insensitive liposomes against HeLa, A549, or MDA-MB-231 MCS.

## 3. Discussion

Two decades of investigations on nano-drug delivery systems have established the importance of their physicochemical properties in targeting the payload drug to cancer cells (also known as physical targeting) [37]. Such physicochemical properties include size, shape, surface charge, surface hydrophilicity, drug-loading, and drug release [37]. The introduction of new characteristics such as active targeting or pH-sensitivity needs to be accomplished in coordination with such properties, which poses a considerable challenge to formulation development.

In this study on ICL, the three imidazole-based lipids triggered PEGylated liposomes by efficiently clustering with phospholipid-PEG conjugates. Such a feature differentiates them from the imidazole lipid reported by Ju et al. [24] and represents a novel approach to construct stealth liposomes with pH-sensitivity. The clustering action is most probably achieved by the three lipids’ unique structure, in which the imidazole headgroup is linked to the lipid tail at the C2 position through a carbon-sulfur bond so that both nitrogen atoms of the imidazole group can serve as H-bond donors upon protonation at acidic pH (Figure 2). The protonated imidazole groups can then each bind with negatively charged phosphate groups from two different DPPE-PEG molecules, which in turn crosslink DPPE-PEG molecules into clusters on the ICL surface. As PEGylation serves as a key method to construct long-circulating liposomes for anticancer drug delivery by the EPR effect, the imidazole-based lipids under this study have the potential for wide applications in vivo.

Compared to the doxorubicin-loaded PEG liposomes (Doxil**^®^**) that are in current clinical use, ICL carries the advantage of pH-sensitivity while preserving the physicochemical properties that favor passive targeting to solid tumors. The drug-free ICL carried sizes under 130 nm in diameter while the DOX-loaded ICL formulations carried sizes under or around 200 nm, both of which were within the size range for the EPR effect [9]. The increase of size and PDI of ICL upon DOX-loading was probably due to the aggregation of DOX molecules with the liposomes because our attempts to load higher concentrations of DOX led to precipitation and because DOX had been reported to aggregate with negatively charged liposomes [38]. DOX can be loaded into ICL at >50% encapsulation efficiency (EE) and at sufficiently high concentrations (Table 1) for anticancer studies in cell culture [39]. The payload DOX concentration of ICL could be further elevated by concentrating DOX-loaded ICL using Tangential Flow Filtration [40]. Although DOX is elected as the cargo drug in this study for better comparison between ICL and clinically established liposomal formulations, we anticipate that the imidazole lipids under this study can be used to trigger PEGylated liposomes containing various water-soluble anticancer drugs.

In response to the drop of pH, ICL without cholesterol demonstrated a number of substantial changes in their physicochemical properties, including acquisition of positive surface charges (ζ-potential elevation in Figure 4A), lipid phase separation (DSC studies in Figure 8A and TEM images in Figure 7), binding with the bio-mimetic membrane (aggregation with model liposomes in Figure 5A), and enhanced release of the payload drug DOX (Figure 6). The extent of most of the changes, namely the positive surface charge acquisition, lipid phase separation, and binding with bio-mimetic membrane, are correlated with higher basicity of the imidazole lipid (DHDMI > DHMI >DHI). This correlation can be explained by our proposed mechanism of ICL’s pH-sensitivity, where the more basic imidazole lipid would be protonated more at the same mildly acidic pH, which would yield more positive charges on liposome surface and more electrostatic interaction between the imidazole lipid and DPPE-PEG, which would, in turn, promote phase separation of ICL membranes and binding between ICL and bio-mimetic membranes. Interestingly, DOX release from ICL did not follow such correlation in that only ICL consisting of DHMI showed substantially enhanced DOX release when the pH dropped from 7.4 to 6.0. TEM images of DHMI/DSPC/DPPE-PEG in comparison to DHI/DSPC/DPPE-PEG and DHDMI/DSPC/DPPE-PEG suggest that this pH-triggered release may be caused by DHMI/DSPC/DPPE-PEG’s unique tendency to collapse into non-lamellar structures at pH 6.0. Alternatively, DHMI/DSPC/DPPE-PEG’s membrane might also have more structural defects at the edge between the separated lipid phases at pH 6.0 to enhance the DOX release.

The incorporation of 25 mol% cholesterol prevented the size increase of ICL during DOX-loading; it also elevated the EE and the payload DOX concentration in ICL. This was probably because cholesterol can improve the stability of the lipid bilayer structure in the ICL formulations. During drug-loading, when the temperature is above the lipid bilayer transition temperature (T > T_m_), the liposome membrane was in the fluid phase, in which the lipid molecules were free to move laterally. The addition of cholesterol was found to help suppress the mobility of lipid bilayers in the fluid phase and reduce their permeability to water, thus improving the membrane stability and drug retention during drug loading [41]. The introduction of cholesterol also diminished the pH-sensitivity of ICL (Figure 4, Figure 5, Figure 6 and Figure 7), probably because the incorporation of cholesterol obstructed the lateral movements of lipids in bilayers at T < T_m_. The nonpolar cholesterol molecules were found to tie up the neighboring lipid’s hydrocarbon chains to minimize cholesterol molecules’ thermodynamically unfavorable exposure to water at the membrane-water interface [42]. In ICL, such cholesterol-lipid interaction would substantially limit the movement of the imidazole-based lipids and DPPE-PEG, thus hindering their clustering at acidic pH. Furthermore, ICL with cholesterol maintained negative ζ-potentials at acidic pH, indicating that the protonation of the imidazole-based lipids was also suppressed by cholesterol (Figure 4B). This is probably because the addition of cholesterol increases the hydrophobicity of the liposome membrane, which in turn reduces its affinity with cations [43].

The targeting of cytotoxic chemotherapy drugs to enhance their efficacy and safety is a complicated process with multiple challenges that all need to be addressed by the drug delivery system, including preferred distribution of the drug molecules from blood circulation to the tumor interstitium, sufficient permeation of the drug molecules to all areas of the solid tumor, and uptake of the drug molecules by virtually all the cancer cells in the tumor. It is therefore critical that anticancer drug delivery systems are evaluated by biological models that simulate these multiple challenges. ICL formulations under this study were evaluated by both monolayer cancer cells and 3D MCS in culture in order to test the potential of their pH-sensitivity to enhance the anticancer activity [34].

In monolayer Hela cells, ICL’s higher ability to suppress cell viability is strongly correlated with lower pH and higher pKa, which are both correlated with the acquisition of positive charges on the ICL surface, ICL’s phase separation, and ICL’s interaction with negatively charged bio-mimetic membranes. Such strong correlations suggest that ICL’s activities to suppress monolayer cell viabilities can be attributed to ICL’s enhanced binding to the cancer cells at lowered pH. More specifically, the drop of pH protonates the imidazole lipids, which cluster DPPE-PEG lipids to expose a de-PEGylated and positively charged ICL surface, which in turn binds to the cancer cell surface to induce the endocytosis of the DOX-loaded ICL and consequently the cell death.

The ICL’s pattern of suppressing the cell viability in 3D multicellular spheroids was quite different from that in 2D monolayer cells. Overall, ICL consisting of DHMI, the imidazole lipid of the second-highest calculated pKa (6.20 ± 0.5), yielded the highest activity to suppress MCS viability, rather than DHDMI of the highest calculated pKa (6.75 ± 0.5). This is probably due to the dynamic balance between the binding of ICL to the cancer cells in MCS and the penetration of ICL to reach the most cancer cells in MCS. On the one hand, DHI may carry too low a pKa (5.53 ± 0.5) to sufficiently trigger its ICL in the mildly acidic microenvironment of MCS; on the other hand, DHDMI may carry too high a pKa, which would trigger most of its ICL to bind to only the cancer cells in the peripheral region of MCS. Paradoxically, DHMI may carry the optimal basicity (pKa 6.20 ± 0.5, closest to the measured interstitial pH of MCS) to facilitate both the penetration and the cellular binding of its ICL. Furthermore, among all the ICL under this study, DHMI/DSPC/DPPE-PEG showed the unique property of pH-enhanced drug release, which would allow such ICL to selectively release DOX in the MCS interstitium to kill multiple adjacent cancer cells, also known as the bystander effect [44].

## 4. Materials and Methods

### 4.1. Materials

1,2-Di-O-hexadecyl-*rac*-glycerol (DHG), 2-mercaptoimidazole, 4-methyl-1H-imidazole-2-thiol, and 4,5-dimethyl-1H-imidazole-2-thiol were purchased from Santa Cruz Biotechnology (Dallas, TX, USA). *p*-Toluenesulfonyl chloride, 2-[4-(2-hydroxyethyl)piperazin-1-yl]-ethanesulfonic acid (HEPES), and 2-(N-morpholino)ethanesulfonic acid (MES) were purchased from Fisher Scientific (Hampton, NH, USA). Triethylamine (TEA) was purchased from Alfa Aesar (Haverhill, MA, USA). The lipids 1,2-distearoyl-*sn*-glycero-3-phosphocholine (DSPC), 1,2-dipalmitoyl-*sn*-glycero-3-phosphoethanolamine-*N*-[azido(polyethylene glycol)-2000 (DPPE-PEG (2000)), 1-palmitoyl-2-oleoyl-*sn*-glycero-3-phosphocholine (POPC), 1-palmitoyl-2-oleoyl-*sn*-glycero-3-phosphoethanolamine (POPE), 1-palmitoyl-2-oleoyl-*sn*-glycero-3-phospho-L-serine (sodium salt) (POPS) and L-α-phosphatidylinositol (Soy) (L-R-PI) were purchased from Avanti Polar Lipids, Inc. (Alabaster, AL, USA). Cholesterol, Dowex^®^ 50WX-4 (50–100 mesh), Sephadex G-25, and Uranyl acetate (UA) were purchased from Sigma-Aldrich (St. Louis, MO, USA). Doxorubicin hydrochloride was purchased from Biotang (Waltham, MA, USA). Carbon-coated copper grids (200 mesh) for electron microscopy were purchased from Polysciences (Warrington, PA, USA). The HeLa, A549, MDA-MB-231 and MDA-MB-468 cell lines were purchased from ATCC (Manassas, VA, USA). The Dulbecco’s Modified Eagle’s Medium (DMEM), Advanced DMEM/F12 medium, Trypsin-EDTA, L-glutamine, fetal bovine serum, and collagen were purchased from Thermo-Fisher Scientific (Waltham, MA, USA). The RPMI 1640 medium, penicillin-streptomycin, 96-well Ultra-low Attachment round-button microplates, 96-well solid white microplates, CellTiter-Glo 3D cell viability assay kits and MTS CellTiter 96^®^ AQueous One Solution cell proliferation assay kits (Promega Corp., WI, USA) were purchased from VWR (Radnor, PA, USA). All other organic solvents and chemicals were purchased from Sigma-Aldrich (St. Louis, MO, USA), Fisher Scientific (Hampton, NH, USA) or VWR (Radnor, PA, USA).

### 4.2. Synthesis of 2,3-di-O-Hexadecyl-1-rac-glyceryl-tosylate (DHG-Tosylate)

1,2-Di-O-hexadecyl-*rac*-glycerol (DHG) (2.30 g, 4.25 mmol) was mixed with anhydrous dichloromethane (20 mL) and pyridine (18.6 mL, 225 mmol). *p*-Toluenesulfonyl chloride (1.90 g, 9.97 mmol) was dissolved in ~0.5 mL anhydrous dichloromethane and transferred into the mixture. The reaction mixture was stirred under argon at room temperature for 8 to 12 h. The reaction mixture was then mixed well with 10 mL anhydrous dichloromethane and washed with saturated Na_2_CO_2_ solution 3 times. The organic phase was separated from the aqueous phase, dried with MgSO_4_, filtered, and then evaporated into dryness under vacuum. The resultant residue was separated by silica gel chromatography with dichloromethane as the mobile phase to yield 2.53 g solid (86%). DART Mass Spectrum: 695.5; calculated, 695.6 (MH)^+^. 1H-NMR (600 MHz, CDCl_3_, δ ppm): 0.87 (t, 6H, 2 C*H*_3_(CH_2_)_15_-), 1.18–1.31 (m, 52H, 2 OCH_2_CH_2_(C*H*_2_)_13_CH_3_), 1.46 (m, 4H, 2 OCH_2_C*H*_2_(CH_2_)_13_CH_3_), 2.44 (s, 3H, -(C_6_H_4_)C*H*_3_), 3.31–3.62 (m, 7H, glyceryl/hexadecyl -C*H*_2_O and -C*H*O-), δ 4.14 (m, 2H, -C*H*_2_OSO_2_-), δ 7.33 (d, 2H, aromatic protons ortho to -CH_3_), and δ 7.78 (d, 2H, aromatic protons ortho to -SO_2_-).

### 4.3. Synthesis of sn-2-((2,3-Dihexadecyloxypropyl)thio)-1H-imidazole (DHI)

2-Mercaptoimidazole (0.91 g, 9.06 mmol) was dissolved in 8–9 mL of anhydrous N, N-dimethylformamide (DMF). DHG-tosylate (1.265 g, 1.82 mmol) was dissolved in 7–8 mL of anhydrous dichloromethane and transferred into the above-mentioned solution, followed by the addition of triethylamine (TEA, 1.27 mL, 9.08 mmol). The reaction mixture was stirred under argon at 55 °C for 48 h. The solvent was evaporated under a vacuum, and the resultant residue was dissolved in dichloromethane. The solution was washed with saturated sodium bicarbonate solution 3 times, dried with sodium carbonate, filtered, and then evaporated into dryness under vacuum. The resultant residue was then separated by silica gel chromatography with 1–5 vol% methanol in dichloromethane as the mobile gradient phase to yield DHI (25–30%). DART Mass Spectrum: 623.48 (Appendix A); calculated, 623.55 (MH)^+^. 1H-NMR (Appendix A, 600 MHz, CDCl_3_): δ 0.87 (t, 6H, 2 C*H*_3_(CH_2_)_15_-), δ 1.19–1.32 (m, 54H, 2 -OCH_2_CH_2_(C*H*_2_)_13_CH_3_ and -*H*_2_CSCNH-), δ 1.55 (m, 4H, 2 OCH_2_C*H*_2_(CH_2_)_13_CH_3_, δ 3.2–3.7 (m, 7H, glyceryl/hexadecyl -C*H*_2_O and -C*H*O-), δ 7.02 (d, 1H, H_2_CSC-NHC*H*=CH-N=), δ 7.21 (d, 1H, H_2_CSC-NHCH=C*H*-N=). Elemental analysis: C 73.27%, H 12.25%, N 4.56%; calculated: C 73.25%, H 11.97%, N 4.50%. Calculated pK_a_ using ACD/pKa DB software: 5.53 ± 0.5.

### 4.4. Synthesis of sn-2-((2,3-Dihexadecyloxypropyl)thio)-5-methyl-1H-imidazole (DHMI)

4-Methyl-1H-imidazole-2-thiol (1.03 g, 9.03 mmol) was used to prepare DHMI, using the same synthesis method as DHI. Yield: 25–30%. DART Mass Spectrum: 637.55 (Appendix A); calculated, 637.57 (MH)^+^. 1H-NMR (Appendix A, 600 MHz, CDCl_3_): δ 0.87 (t, 6H, 2 C*H*_3_(CH_2_)_15_-), δ 1.19–1.34 (m, 54H, 2 -OCH_2_CH_2_(C*H*_2_)_13_CH_3_ and -*H*_2_CSCNH-), δ 1.53 (m, 4H, 2 OCH_2_C*H*_2_(CH_2_)_13_CH_3_), δ 2.41 (s, 3H, -H_2_CSC-NH-C(C*H*_3_)-), δ 3.2–3.7 (m, 7H, glyceryl/hexadecyl -C*H*_2_O and -C*H*O-), δ 6.81 (s, 1H, -H_2_CSC=N-C*H*=). Elemental analysis: C 73.63%, H 12.08%, N 4.35%; calculated: C 73.52%, H 12.02%, N 4.40%. Calculated pK_a_ using ACD/pKa DB software: 6.20 ± 0.5.

### 4.5. Synthesis of sn-2-((2,3-Dihexadecyloxypropyl)thio)-4,5-dimethyl-1H-imidazole (DHDMI)

4,5-Dimethyl-1H-imidazole-2-thiol (1.15 g, 9.03 mmol) was used to prepare DHDMI, using the same synthesis method as DHI. Yield: 25–30%. DART Mass Spectrum: 651.56 (Appendix A); calculated, 651.59 (MH)^+^. 1H-NMR (Appendix A, 600 MHz, CDCl_3_): δ 0.87 (t, 6H, 2 C*H*_3_(CH_2_)_15_–), δ 1.19–1.34 (m, 54H, 2 -OCH_2_CH_2_(C*H*_2_)_13_CH_3_ and -*H*_2_CSCNH-), δ 1.53 (m, 4H, 2 OCH_2_C*H*_2_(CH_2_)_13_CH_3_), δ2.22 (s, 3H, -H_2_CSC-NHC(C*H*_3_)=), δ2.24 (s, 3H, -H_2_CSC=N-C(C*H*_3_)=), δ 3.3–3.7 (m, 7H, glyceryl/hexadecyl -C*H*_2_O and -C*H*O-). Elemental analysis: C 73.78%, H 12.24%, N 4.16%; calculated: C 73.78%, H 12.07%, N 4.30%. Calculated pK_a_ using ACD/pKa DB software: 6.72 ± 0.5.

### 4.6. Preparation of ICL Formulations

A dichloromethane solution of an imidazole-based lipid and a chloroform solution of other lipids were mixed in a round-bottom flask. The organic solvents were evaporated under reduced pressure to form a lipidic film at 70 °C. The lipidic film was further dried in a high vacuum for over 4 h at room temperature to remove the residual solvent. The lipidic film was then hydrated with HEPES buffer (pH 7.4, 30 mM HEPES) containing 300 mM MnSO_4_ by intermittent agitation in a 70 °C water bath under argon to obtain a liposome suspension containing 20 mM total lipids. The liposome suspension was freeze-anneal-thawed by rapidly freezing in liquid nitrogen, immerging in the ice-water mixture for 2 min and incubating in a 70 °C water bath for 4 min. The freeze-anneal-thawing was repeated 11 times. The liposome suspension was sequentially extruded 21 times each through 400 nm, 200 nm and 100 nm polycarbonate membranes (Nucleopore Corp., Pleasanton, CA, USA) using a hand-held Mini-extruder (Avanti Polar Lipids Inc., Alabaster, AL, USA) at 70 °C to reduce and homogenize the size of liposomes. DOX was then loaded into the liposomes as follows, using a transmembrane MnSO_4_ gradient [45]. The extruded liposomes were separated from the unencapsulated MnSO4 by size exclusion chromatography using a Sephadex G-75 column pre-equilibrated with isotonic HEPES buffer (pH 7.4, 5 mM HEPES, 140 mM NaCl). DOX (0.75 mg/mL) dissolved in the same isotonic HEPES buffer was then mixed with the liposome suspension in a 1:2 (*v*/*v*) ratio, and the mixture was incubated in a 70 °C water bath for 90 min. The cation-exchange resin Dowex^®^ 50WX-4, 50–100 mesh was pre-treated with NaOH and NaCl [46], mixed with the DOX-liposome mixture at DOX: resin = 1:60 (*w*/*w*), and then shaken gently for 25 min to remove the unencapsulated DOX from the DOX-loaded liposomes. The resin was separated from the DOX-loaded liposomes by filtration. A tangential flow filtration column (MicroKros^®^, Spectrum, Stamford, CT, USA) was used to concentrate the liposome suspension by partially removing the extra-liposomal buffer. Typically, a 2 mL liposome suspension was extruded 14 times through the tangential flow filtration column to yield a ~0.5 mL concentrated formulation. The lipid composition of the liposomes under study is listed in Table 3.

### 4.7. Size Measurement

An aliquot (2.5–5 μL) of a liposome suspension was diluted in 150 μL isotonic buffer, and the size was measured at room temperature by dynamic light scattering (Zetasizer ZS90, Malvern Instruments Ltd., Malvern, Malvern, UK). The data were analyzed based on light intensity distribution to give hydrodynamic diameters.

### 4.8. Quantification of Encapsulation Efficiency (EE)

An aliquot (10 μL) of DOX-loaded liposome suspension was lysed with 90 μL lysis buffer (90% (*v*/*v*) isopropanol, 0.075 M HCl) [46] in a 96-well Black Clear Bottom Polystyrene microplate (Corning^®^, NY, USA), together with 10 μL DOX standard solutions diluted in the same lysing buffer (90 μL). The microplate was covered with foil, and the fluorescence of the samples (λ_ex_ = 486 nm, λ_em_ = 590 nm) was recorded on a Synergy HT microplate reader (Biotek, Winooski, VT, USA). The concentration of the payload DOX of liposomes was estimated using a standard calibration curve from the fluorescence of the DOX standard solutions. The encapsulation efficiency (EE) of the liposomes was then calculated by the following formula.
$$\mathrm{EE}=\frac{\mathrm{Encapsulated}\;\mathrm{DOX}\;\mathrm{Conc}.\;}{\mathrm{Input}\;\mathrm{DOX}\;\mathrm{Conc}.\;\mathrm{for}\;\mathrm{drug}\;\mathrm{loading}}\times 100\%$$

### 4.9. Differential Scanning Calorimetry

A VP-DSC Instrument (MicroCal, LLC, Northampton, MA, USA) was used for the differential scanning calorimetry (DSC) studies. DSC scans were performed on 0.5 mL liposome suspensions containing 2.5 mM total lipids at pH 7.4 and pH 6.0. The thermograms of liposome suspensions were acquired from 40 °C to 75 °C at a scan rate of 5 °C/h. Each excess heat capacity curve of a liposome sample was normalized by subtraction of the thermogram of the buffer acquired simultaneously under identical conditions.

### 4.10. pH-Triggered Change of ζ-Potential

In order to enhance the detection of changes in liposome surface charge, the liposomes were prepared by hydration in an isotonic buffer of low ionic strength (pH 7.4, 5 mM HEPES, 5% (*w*/*v*) Glucose) [47]. Aliquots (50–100 μL) of the resultant liposome suspensions were diluted in 900 μL isotonic MES buffer (final pH 6.0 and 6.5, 10 mM MES, 5% (*w*/*v*) Glucose) and 900 μL isotonic HEPES buffer (final pH 7.0 and 7.4, 10 mM HEPES, 5% (*w*/*v*) Glucose). The *ζ-*potential was then measured at 37 °C based on electrophoresis mobility under applied voltage (Zetasizer ZS90, Malvern Instruments Ltd., Malvern, UK).

### 4.11. pH-Dependent Interaction with Model Liposomes

The model liposomes (POPC:POPE:POPS:L-R-PI:cholesterol = 50:20:5:10:15 (mol%)) mimicking the lipid composition and surface charge of biomembranes were prepared based on a previous report [31]. As measured by Zetasizer ZS90, the mean size of the model liposomes was 192.7 nm in diameter, and the *ζ-*potential was −51.77 ± 1.18 mV. Suspensions of ICL and pH-insensitive control liposomes were each mixed with the model liposomes at a 1:1 total lipid molar ratio. An aliquot (5 μL) of each mixture was diluted in 150 μL isotonic MES buffer (final pH 6.0 and 6.5, 10 mM MES, 140 mM NaCl) and isotonic HEPES buffer (final pH 7.0 and 7.4, 10 mM HEPES, 140 mM NaCl), and incubated at 37 °C for 5 min. The particle size of the diluted mixtures was measured at 37 °C by dynamic light scattering (Zetasizer ZS90, Malvern Instruments Ltd., UK).

### 4.12. pH-Dependent Drug Release

Each liposome formulation was severally diluted (100 μL aliquots) with 500 μL MES buffer (final pH 6.0 and 6.5, 100 mM MES, 1.7% (*w*/*v*) Glucose) and HEPES buffer (final pH 7.0 and 7.4, 100 mM HEPES, 1.7% (*w*/*v*) Glucose). An aliquot (10 μL) of each diluted liposome formulation was immediately lysed with 90 μL lysis buffer (90% (*v*/*v*) isopropanol, 0.075 M HCl) in a 96-well Black Clear Bottom Polystyrene microplate and the initial DOX concentration *C_i_* (as at the 0-h time point) was qualified with standard DOX solutions. The liposome samples diluted by buffers at various pH (6.0, 6.5, 7.0, 7.4) were then mixed with cation-exchange resin Dowex^®^ 50WX-4 (50–100 mesh) at DOX:resin = 1:200 (*w*/*w*) ratio. The mixtures were incubated and gently shaken at 37 °C. At different time points (1, 3, 6, 12 h), each mixture was allowed to settle briefly, and an aliquot (10 μL) of the resultant supernatant was harvested, lysed, and its DOX concentration measured as mentioned before. The percentage of DOX release was determined by the following equation,
$$\%\;\mathrm{Release}=\left(1-\frac{C_{s}}{C_{i}}\right)\times 100\%$$
where C_s_ is the concentration of DOX in the supernatant of the liposome-resin mixture at different time points, C_i_ is the initial liposomal DOX concentration.

### 4.13. Transmission Electron Microscopy

The morphology of ICL formulations was observed on a JEOL-JEM 1230 Electron Microscope (JEOL, Tokyo, Japan). Carbon-coated copper TEM grids (200 mesh) were subjected to glow discharge before usage to increase their hydrophilicity. An aliquot (5 µL) of diluted ICL suspension (approximately 1 mM total lipids) at pH 7.4 or 6.0 was dripped onto the grid to wet its surface for 1 min and then blotted with filter paper to generate a thin film. The sample film was then wetted five times with 5 µL of the negative stain 2% uranyl acetate (UA) between blotting. The grid was dried at room temperature and then transferred into the electron microscope for imaging at an accelerating voltage of 100 kV. The samples of ICL mixed with model liposomes were prepared and imaged by the same method.

### 4.14. Cell Culture

Cervical cancer cell line HeLa, lung cancer cell line A549, and breast cancer cell lines MDA-MB-231 and MDA-MB-468 were cultured to construct 3D MCS in order to evaluate the anticancer activities of ICL. Hela cells were also cultured into monolayer cells. HeLa cells were maintained in DMEM media; A549 cells were maintained in RPMI 1640 media; MDA-MB-231 and MDA-MB-468 cells were maintained in advanced DMEM/F12 media. All media were supplemented with 10% fetal bovine serum, 1% Penicillin-Streptomycin, and 1% L-glutamine. All cells were grown in a humidified atmosphere of 5% CO_2_ in air at 37 °C and passaged at 85% confluence. In all studies, the cells were sub-cultured every 2–3 days and used for experiments at passages 5–20.

### 4.15. Cytotoxicity Assays on 2D Monolayer Hela Cells

Monolayer HeLa cells at ~85% confluence were suspended by trypsinization, and the cell density was determined with a Handheld Automated Cell Counter (Millipore, Burlington, MA, USA). The cells were then diluted to ~80,000 cells/mL in complete growth media and seeded into 96-well Clear Microplates (Corning, NY, USA) at ~8000 cells/well by transferring 100 μL of the cell suspension into each well. The cytotoxicity assay was carried out on the cells 8 h after they were seeded. The cells were washed with PBS and treated with DOX-loaded liposomes or free DOX solutions in complete growth media at incremental concentrations. The pH of the media (10 mL) was adjusted to 7.4, 7.0, 6.5, and 6.0 with glacial acetic acid. After 12-h incubation, the media was removed, and the cells were washed with 100 μL PBS buffer and supplemented with 100 μL/well complete growth media and 20 µL/well MTS CellTiter 96^®^ AQueous One Solution. The mixture was incubated for 4 h at 37 °C with 5% CO_2_. The cell viability was quantified by UV/visible absorbance at 490 nm on a Synergy HT microplate reader (Biotek, Winooski, VT, USA). The Hela cells treated with growth media at corresponding pHs without free DOX or DOX-loaded liposomes were referred to as 100% cell viability.

### 4.16. Cytotoxicity Assays on 3D Multicellular Spheroids

Monolayer cells in T75 flasks were trypsinized, and the cell density in the suspensions was determined with a Handheld Automated Cell Counter. The cells were then seeded into 96-well Ultra-low Attachment (ULA) round-bottom microplates (Corning, NY, USA) at ~500 Hela cells/well, ~5000 A549 cells/well, ~3000 MDA-MB-231 cells/well, and ~2000 MDA-MB-468 cells/well by transferring 100 μL properly diluted cell suspensions in complete growth media containing collagen (0% for HeLa, 0.3% for A549, 1% for MDA-MB-231, and 1% for MDA-MB-468 cell lines). If needed, the microplates were centrifuged at 7 °C to promote cell aggregation (Appendix A). Complete growth media (100 μL) was added to each well on the second day after seeding. The growth media were then partially exchanged every other day by replacing 100 μL of media in each well with 100 μL fresh media to maintain a 200 μL/well total media volume. The cytotoxicity assays were carried out on selected MCS whose diameter reached or exceeded 500 μm (Appendix A) [48,49]. Part of the growth media (100 μL/well) was replaced with the same volume of DOX-loaded liposomes or free DOX solutions in complete growth media at incremental concentrations. After 72 h incubation, each MCS was transferred into a well of a 96-well Solid White microplate (Corning, NY, USA) together with 100 μL media. Then, 100 μL reagent of the CellTiter-Glo 3D cell viability assay was then added to each well, and the microplate was covered with foil, shaken on an orbital shaker for 5 min, and then incubated for 25 min at room temperature. The viability of MCS was then measured by luminescence intensity on a Synergy HT microplate reader (Biotek, Winooski, VT, USA). The MCS treated by growth media without free DOX or DOX-loaded liposomes was referred to as 100% cell viability.

## 5. Conclusions

Novel imidazole-based convertible liposomes (ICL) have been designed and constructed. ICL carries a PEG-coating and slight excess of negative surface charges at pH 7.4. As pH decreased to 6.0, the imidazole-based lipids assumed positive charges and clustered with negatively charged PE-PEG conjugates in ICL, which in turn partially de-PEGylated the liposomes to enhance their adsorption to negatively charged, bio-mimetic membranes. The drop of pH to 6.0 also enhanced the release of the anticancer drug DOX from ICL that consisted of the imidazole lipid DHMI (>50% release in 6 h), but not those of the other two imidazole-based lipids. TEM studies suggest that DHMI enhanced the drug release from ICL due to its ability to convert the liposomal membrane into non-lamellar structures. The incorporation of cholesterol improved the colloidal stability of ICL but diminished their pH-sensitivity. ICL demonstrated substantially higher anticancer activities than the analogous PEGylated, pH-insensitive liposomes containing doxorubicin, which is a common type of nano-formulations in clinical use. While the anticancer activities of ICL against monolayer Hela cells are correlated with higher pKa of the imidazole lipid, the anticancer activities against 3D multicellular spheroids are the highest in ICL that consisted of the imidazole lipid DHMI, which possesses the medium pKa and enhances the liposomal drug release at pH 6.0. Our studies on ICL suggest that nano-drug delivery systems that balance the needs of intratumoral penetration, adsorption to cancer cells, and enhanced drug release would yield optimal anticancer activities.

## Figures and Tables

**Figure 1 pharmaceuticals-15-00306-f001:**
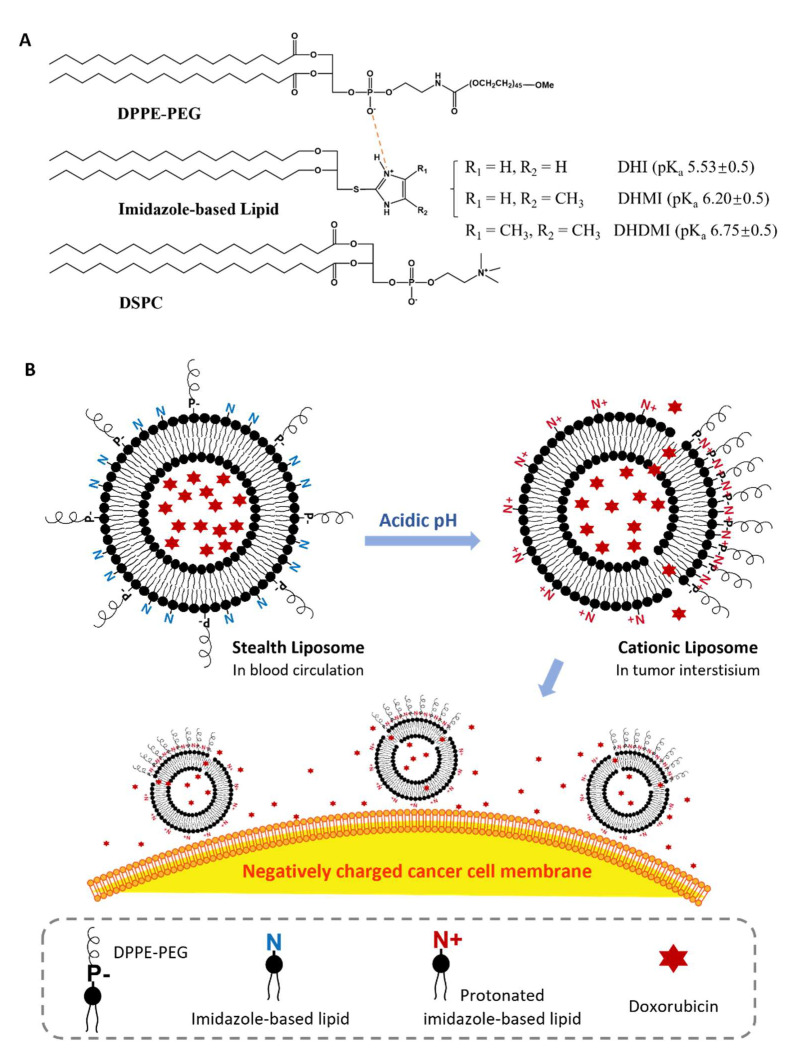
Imidazole-based convertible liposome (ICL). (**A**) Chemical structures of lipids that constitute ICL. (**B**) Schematic of ICL turning from stealth liposomes into cationic liposomes in acidic tumor interstitium. P-, negatively charged phosphate group in DPPE-PEG; N/N+, basic amine in imidazole-based lipids.

**Figure 2 pharmaceuticals-15-00306-f002:**
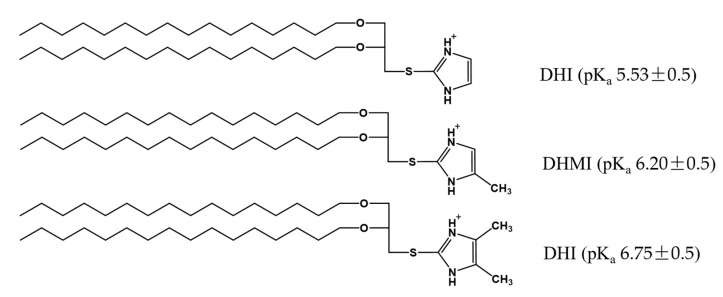
Imidazole-based, pH-sensitive lipids under study: chemical structures and calculated pKa by the ACD/pKa DB software.

**Figure 3 pharmaceuticals-15-00306-f003:**
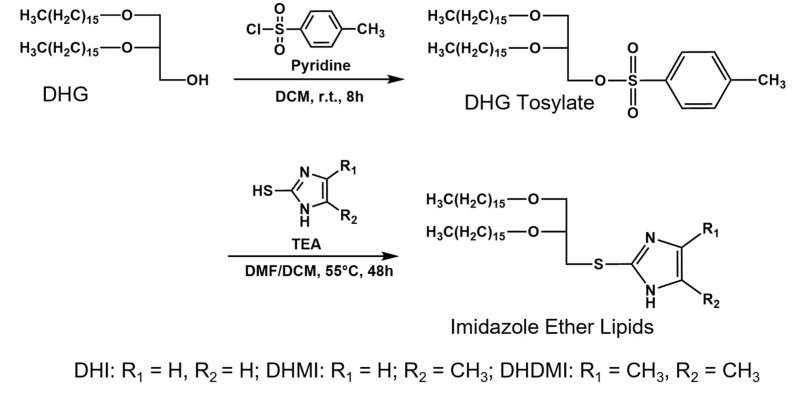
Synthesis of imidazole-based, pH-sensitive lipids.

**Figure 4 pharmaceuticals-15-00306-f004:**
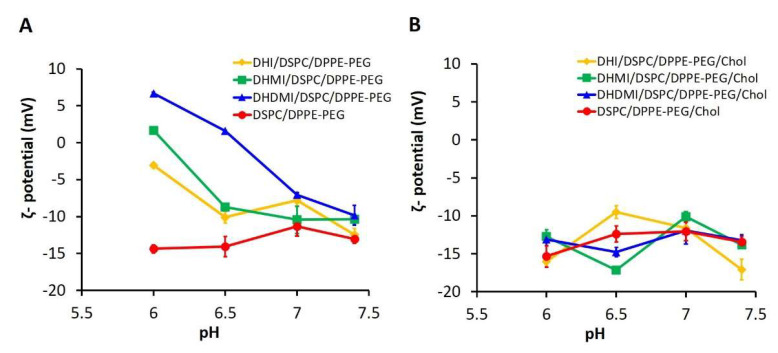
pH-triggered acquisition of positive surface charges by ICL. ζ-potential of ICL without (**A**) or with (**B**) 25% cholesterol were measured at 37 °C, pH 6.0, 6.5, 7.0, and 7.4. Data are presented as mean ± SD, *N* = 3.

**Figure 5 pharmaceuticals-15-00306-f005:**
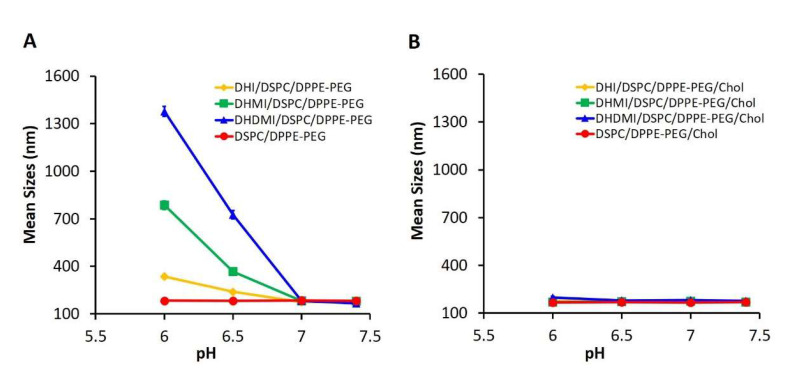
pH-triggered interaction between ICL and model liposomes. Mean sizes of equimolar mixture of model liposome and ICL without (**A**) or with (**B**) 25% cholesterol were measured at 37 °C, pH 6.0, 6.5, 7.0, and 7.4. Data are presented as mean ± SD, *N* = 3.

**Figure 6 pharmaceuticals-15-00306-f006:**
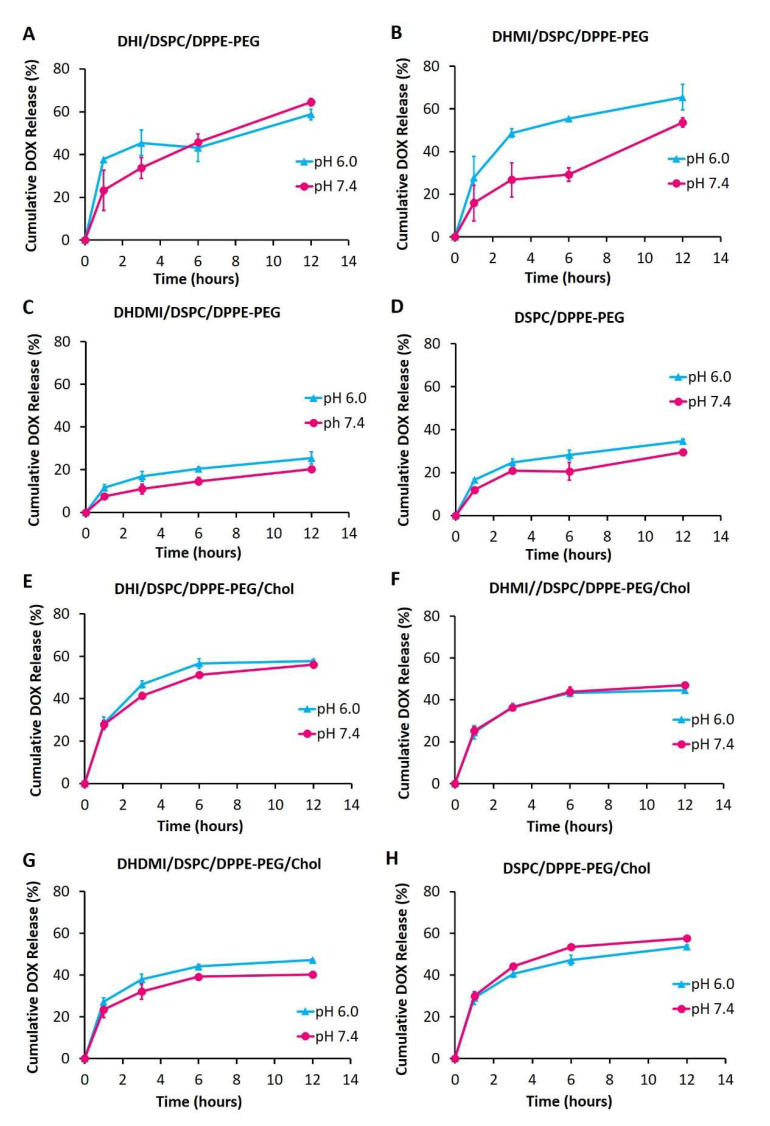
Release of DOX from liposomes without cholesterol (**A**–**D**) and with 25% cholesterol (**E**–**H**) over 12 h of incubation at 37 °C, pH 6.0 and 7.4. Data are presented as mean ± SD, *N* = 3.

**Figure 7 pharmaceuticals-15-00306-f007:**
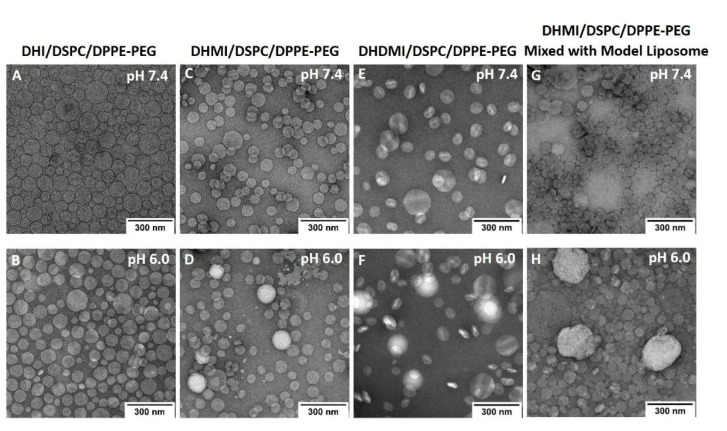
TEM images of DOX-loaded ICL formulations (**A**–**F**) and mixture of DHMI liposomes with model liposomes (**G**,**H**) at pH 7.4 (**A**,**C**,**E**,**G**) and pH 6.0 (**B**,**D**,**F**,**H**).

**Figure 8 pharmaceuticals-15-00306-f008:**
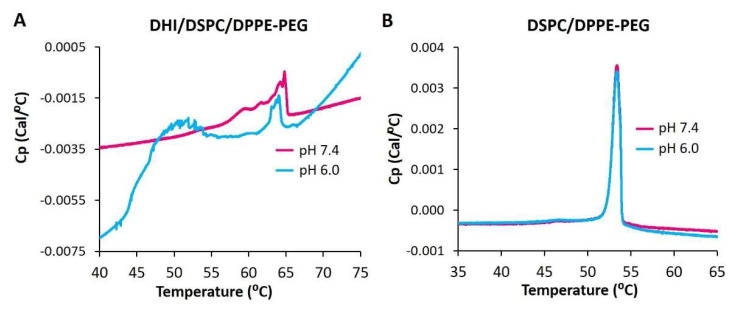
DSC thermograms of pH-sensitive ICL (DHI/DSPC/DPPE-PEG) (**A**) and pH-insensitive stealth liposome DSPC/DPPE-PEG (**B**) at pH 7.4 and 6.0.

**Figure 9 pharmaceuticals-15-00306-f009:**
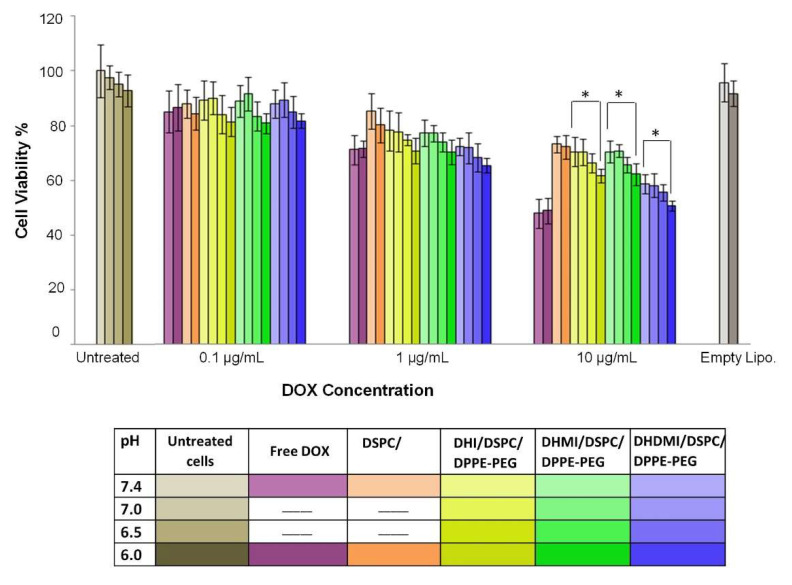
Cell viability of 2D HeLa cells treated with free DOX, cholesterol-free ICL and pH-insensitive liposomes at pH 7.4, 7.0, 6.5, and 6.0 for 12 h. Data are presented as mean ± SD, N = 4. * *p* ˂ 0.05. pH was adjusted in the growth media.

**Figure 10 pharmaceuticals-15-00306-f010:**
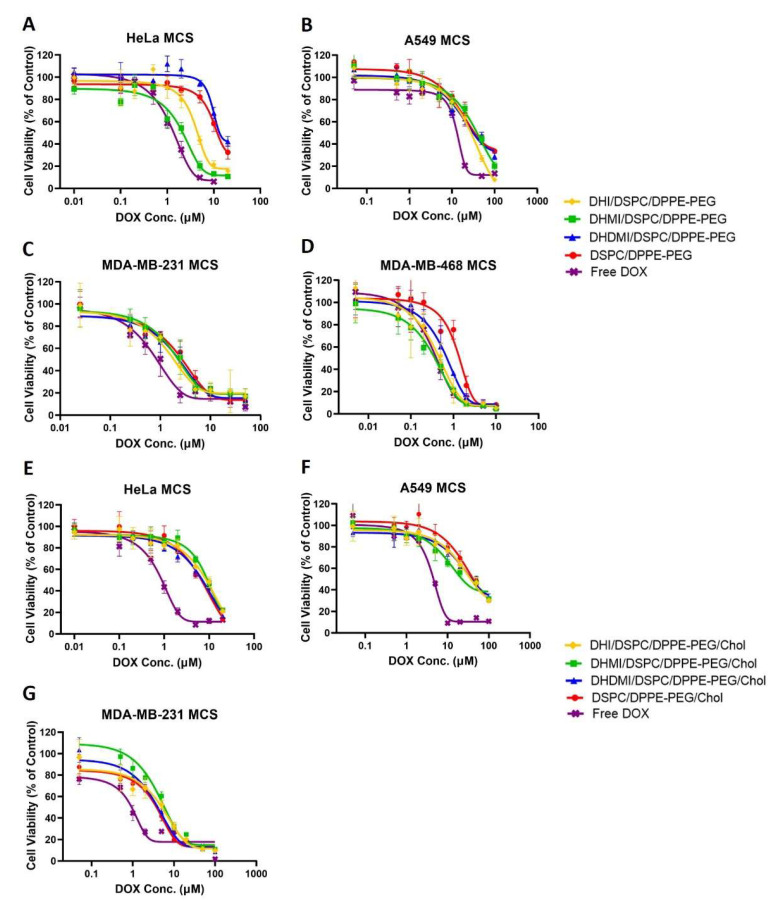
Cell viability of Hela, A549, MDA-MB-231, and MDA-MB-468 MCS after treatment with incremental concentrations of ICL formulations, pH-insensitive liposome, or free DOX. ICL consisted of either no cholesterol (**A**–**D**) or 25 mol% cholesterol (**E**–**G**). Data are presented as mean ± SD, *N* = 4.

**Table 1 pharmaceuticals-15-00306-t001:** Physicochemical characteristics of DOX-free and DOX-loaded ICL in comparison with pH-insensitive stealth liposomes.

		Before DOX-Loading		After DOX-Loading		
Lipid Compositions	Molar Ratio	Size (nm)	PDI	Size (nm)	PDI	EE (%)
DHI/DSPC/DPPE-PEG	25/70/5	114.9 ± 10.9	0.205 ± 0.008	200.8 ± 14.6	0.522 ± 0.047	56.62 ± 2.06
DHMI/DSPC/DPPE-PEG	25/70/5	117.9 ± 3.6	0.220 ± 0.033	189.3 ± 22.3	0.546 ± 0.055	53.18 ± 1.12
DHDMI/DSPC/DPPE-PEG	25/70/5	104.8 ± 3.5	0.176 ± 0.035	194.9 ± 7.0	0.253 ± 0.130	59.54 ± 0.59
DSPC/DPPE-PEG	95/5	101.8 ± 3.0	0.125 ± 0.067	136.9 ± 13.7	0.364 ± 0.085	57.74 ± 0.98
DHI/DSPC/DPPE-PEG/Chol	25/45/5/25	122.7 ± 8.19	0.081 ± 0.020	142.2 ± 9.5	0.113 ± 0.035	71.38 ± 0.61
DHMI/DSPC/DPPE-PEG/Chol	25/45/5/25	116.5 ± 11.9	0.075 ± 0.030	128.1 ± 8.3	0.078 ± 0.021	89.86 ± 1.27
DHDMI/DSPC/DPPE-PEG/Chol	25/45/5/25	114.0 ± 5.6	0.074 ± 0.008	128.6 ± 14.2	0.104 ± 0.020	92.97 ± 1.10
DSPC/DPPE-PEG/Chol	70/5/25	119.9 ± 5.4	0.066 ± 0.037	138.4 ± 5.6	0.165 ± 0.092	60.98 ± 1.66

Size values are hydrodynamic diameters based on cumulative intensity. Data are presented as mean ± SD, *N* = 3.

**Table 2 pharmaceuticals-15-00306-t002:** IC50 values of DOX-loaded liposomes and free DOX on HeLa, A549, MDA-MB-231, and MDA-MB-468 3D MCS.

Liposome Membrane Composition	Lipid Molar Ratio	IC_50_ ^$^ (μM)			
		Hela	A549	MDA- MB-231	MDA- MB-468
DHI/DSPC/DPPE-PEG	25/70/5	3.82 ± 1.13 ***	~30 ^#^	1.38 ± 1.31	0.38 ± 0.21 **
DHMI/DSPC/DPPE-PEG	25/70/5	2.07 ± 1.13 ***	~40 ^#^	1.77 ± 1.21	0.31 ± 0.15 ***
DHDMI/DSPC/DPPE-PEG	25/70/5	9.51 ± 1.15	~35 ^#^	1.86 ± 1.24	0.63 ± 0.10 **
DSPC/DPPE-PEG	95/5	11.41 ± 1.28	~35 ^#^	2.37 ± 1.29	1.24 ± 0.13
DHI/DSPC/DPPE-PEG/Chol	25/45/5/25	~10 ^#^	~30 ^#^	5.13 ± 1.46	-
DHMI/DSPC/DPPE-PEG/Chol	25/45/5/25	10.38 ± 1.33	29.07 ± 2.73	3.62 ± 1.17	-
DHDMI/DSPC/DPPE-PEG/Chol	25/45/5/25	~10 ^#^	24.06 ± 1.40	3.26 ± 1.18	-
DSPC/DPPE-PEG/Chol	70/5/25	~10 ^#^	33.88 ± 1.62	3.98 ± 1.10	-
Free DOX	-	1.26 ± 0.04	12.59 ± 1.05	1.18 ± 0.29	0.32 ± 0.12

^$^ Calculated from dose-response data using GraphPad software. Data are presented as mean ± SD, *N* = 4. ** *p* < 0.01, *** *p* < 0.001 compared to the IC_50_ of DSPC/DPPE-PEG. ^#^ Visual estimation from dose-response curve.

**Table 3 pharmaceuticals-15-00306-t003:** Lipid composition of liposomes under study.

	Mol %					
Formulations	DHI	DHMI	DHDMI	DSPC	DPPE-PEG	Chol
I	25	-	-	70	5	-
II	-	25		70	5	-
III	-	-	25	70	5	-
IV	-	-	-	95	5	-
V	25	-	-	45	5	25
VI	-	25	-	45	5	25
VII	-	-	25	45	5	25
VIII	-	-	-	70	5	25

I–III and V–VII are ICL; IV and VIII are PEGylated, pH-insensitive liposomes as controls; V–VIII contain cholesterol, and I–IV does not contain cholesterol.

## Data Availability

Data is contained within the article and Appendix A.

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
