# Peer review of "Imidazole-Based pH-Sensitive Convertible Liposomes for Anticancer Drug Delivery"

_pharmaceuticals, 2022, doi:10.3390/ph15030306_

Round 1

Reviewer 1 Report

Title : Imidazole-based pH-sensitive Convertible Liposomes for Anti- 2
cancer Drug Delivery

I appreciate the authors for the efforts for the manuscript

It was well designed and written.

Minor suggestions: to improve the worth of the manuscript.

Abstract: Can be improved with presenting exact results in brief in few lines.

Introduction: Is appropriate for this manuscript

What is the reason behind the selection of Doxorubicin? As already Doxorubicin liposomes are present in market?

Are there any advantage by these formulations over marketed formulations? Please justify this.

Materials and methods: Well written

Results and discussion: 

Authors presented the release profiles from different liposomes:

It would be great if authors try to fit into release kinetics models to give worth to the manuscript.

Please Refer: Cabazitaxel-loaded nanocarriers for cancer therapy with reduced side effects : https://doi.org/10.3390/pharmaceutics11030141

Author Response

Reviewer 1

We thank the reviewer for contributing precious time and for providing constructive feedback. The following are our point-by-point response to the reviewer’s suggested modifications.

“Abstract: Can be improved with presenting exact results in brief in few lines.”

The abstract has been shortened to 203 words, which better follows the journal’s guidelines for conciseness. The main results of the paper are summarized in about five lines:” Upon the drop of pH, ICL gained more positive surface charges, displayed lipid phase separation in TEM and DSC, and aggregated with cell membrane-mimetic model liposomes. The drop of pH also enhanced DOX release from ICL consisting of one of the imidazole lipids, sn-2-((2,3-dihexadecyloxypropyl)thio)-5-methyl-1H-imidazole. ICL demonstrated superior activities against monolayer Hela cells and several 3D MCS than the analogous PEGylated, pH-insensitive liposomes containing DOX, which serves as a control and clinical benchmark.”

“What is the reason behind the selection of Doxorubicin? As already Doxorubicin liposomes are present in market?”

The key novelty of this paper is the advantage of the three imidazole lipids, which can provide pH-sensitivity in cooperation with the PEG lipids in stealth liposomes. Doxorubicin is chosen as cargo drugs so that our ICL can have a more direct comparison with PEGylated, pH-insensitive liposomes containing DOX, which serves both as a control, and as a clinical benchmark. We anticipate that ICL can be used to delivery other drugs, depending on the specific anticancer application.

We added the following texts in the manuscript to better explain our novelty and our selection of Doxorubicin, which is not as an optimal payload for anticancer activity, but as a model payload drug that has already demonstrated substantial clinical success:

In the abstract: “ …the analogous PEGylated, pH-insensitive liposomes containing DOX, which serves as a control and clinical benchmark.”

In the introduction: “Herein we report a novel type of imidazole lipids and their pH-sensitive liposomes. Our goal is to develop imidazole lipids that trigger the liposomes in cooperation with phosphatidylethanolamine -polyethylene glycol conjugates (PE-PEG), which is a key component to stabilize liposomes in blood circulation for anticancer drug delivery.”

In discussion: “Although DOX is elected as the cargo drug in this study for better comparison between ICL and clinically established liposomal formulations, we anticipate that the imidazole lipids under this study can be used to trigger PEGylated liposomes containing various water-soluble anticancer drugs.”

“Are there any advantage by these formulations over marketed formulations? Please justify this.”

As stated in our response to the previous comment, we designed our studies so we can show ICL’s advantage over an important group of marketed formulations: PEGylated liposomal formulations of doxorubicin. ICL has demonstrated superior anticancer activities in monolayer Hela cells and, more importantly, in several multicellular spheroids of cancer cell lines over the PEGylated and pH-insensitive liposome formulation of doxorubicin. We have modified the manuscript to better explain such advantage.

“Authors presented the release profiles from different liposomes: It would be great if authors try to fit into release kinetics models to give worth to the manuscript.”

We thank the reviewer for this comment. We have started more comprehensive studies on the release profile of ICL consisted of DHMI, not only in test tube, but also in multicellular spheroids of cancer cells under confocal microscope. We are studying the correlation between the release profile, the drug distribution in MCS, and the anticancer activity of MCS and plan to publish the findings in a separate manuscript.

“Please Refer: Cabazitaxel-loaded nanocarriers for cancer therapy with reduced side effects: https://doi.org/10.3390/pharmaceutics11030141

We thank the reviewer for bringing this paper to our attention. We have included this paper as reference 11 in the manuscript.

Reviewer 2 Report

I found the manuscript interesting due to the perspective of developing imidazole-based lipids and liposomes with nanometric size able to release anticancer drugs to all regions of tumors with acidic microenviroments.

I have some suggestions for improving readability of the manuscript:

1)Line 9: please, define DHI, DHMI and DHDMI or eliminate these abbreviations from the abstract.

2)Line 17: Define DHMI

3)On Fig 1 A, please replace R by R1 or R2 in the chemical structure of the lipid.

4)The best drug release was 60% at 12h incubation achieved for DHMI at pH 6.0. Please, emphasize this in the Conclusions section and explain why this lipid facilitates DOX release.

5) Still in the Conclusions section, emphasize the importance of a moe neutral (less negatively charged liposome to improve adsorption to cancer cells; add also the importance of cholesterol (Fig. 5B) or negative charges (Fig. 5 A) to obtain nanoliposomes.

6) Comment on the zeta-potential values so close to zero at pH6-6.5 (Fig 4A in green and blue); one cannot attribute positive charges to such liposomes; they are basically neutral due to the presence of cationic and anionic lipids in the same bilayer.

Author Response

Reviewer 2:

We thank the reviewer for contributing precious time and for providing constructive feedback. The following are our point-by-point response to the reviewer’s suggested modifications.

1) Line 9: please, define DHI, DHMI and DHDMI or eliminate these abbreviations from the abstract.

We thank the reviewer for this important feedback to improve the clarity of the manuscript. We have eliminated the abbreviations from the abstract. We kept the texts regarding DHMI in the abstract to highlight its advantages in drug release and in anticancer activities but we used its full name in the abstract: sn-2-((2,3-dihexadecyloxypropyl)thio)-5-methyl-1H-imidazole.

We have provided the full names of the three lipids when they are first mentioned in the results section of the manuscript: “Three novel lipids (Figure 2), namely sn-2-((2,3-dihexadecyloxypropyl)thio)-1H-imidazole (DHI), sn-2-((2,3-dihexadecyloxypropyl)thio)-5-methyl-1H-imidazole (DHMI), and sn-2-((2,3-dihexadecyloxypropyl)thio)-4,5-dimethyl-1H-imidazole (DHDMI) were designed as a critical component of ICL.”

We have also provided their full names in the experimental part of the manuscript for clarity.

2) Line 17: Define DHMI

As mentioned in the response to comment 1, we have modified the abstract and the result sections of the manuscript accordingly.

3) On Fig 1 A, please replace R by R1 or R2 in the chemical structure of the lipid.

We have modified Fig 1 A accordingly.

4) The best drug release was 60% at 12h incubation achieved for DHMI at pH 6.0. Please, emphasize this in the Conclusions section and explain why this lipid facilitates DOX release.

We have added the following sentence in Conclusions to highlight the enhanced release of ICL with DHMI at pH 6.0: “The drop of pH to 6.0 also enhanced the release of the anticancer drug DOX from ICL that consisted of the imidazole lipid DHMI (> 50% release in 6 h), but not those of the other two imidazole-based lipids.”

The revised discussion section has the following tests to explain why this lipid could facilitate DOX release: “TEM images of DHMI/DSPC/DPPE-PEG in comparison to DHI/DSPC/DPPE-PEG and DHDMI/DSPC/DPPE-PEG suggest that this pH-triggered release may be caused by DHMI/DSPC/DPPE-PEG’s unique tendency to collapse into non-lamellar structures at pH 6.0. Alternatively, DHMI/DSPC/DPPE-PEG’s membrane might also have more structural defects at the edge between the separated lipid phases at pH 6.0 to enhance the DOX release.”

5) Still in the Conclusions section, emphasize the importance of a more neutral (less negatively charged liposome to improve adsorption to cancer cells; add also the importance of cholesterol (Fig. 5B) or negative charges (Fig. 5 A) to obtain nanoliposomes.

We have added the following sentence in the conclusion section to emphasize this feature of turning slightly negatively charged ICL to neutrally charged or slightly positively charged ICL: “ICL carry a PEG-coating and slight excess of negative surface charges at pH 7.4. As pH decreased to 6.0, the imidazole-based lipids assumed positive charges and clustered with negatively charged PE-PEG conjugates in ICL, which in turn partially de-PEGylated the liposomes to enhance their adsorption to negatively charged, bio-mimetic membranes.”

We have added the following sentence in the modified conclusion section to summarize the role of cholesterol: “The incorporation of cholesterol improved the colloidal stability of ICL but diminished their pH-sensitivity.”

We have also added the following sentence in the abstract to briefly summarize the role of cholesterol: “The presence of cholesterol in ICL enhanced their colloidal stability but diminished their pH-sensitivity.”

6) Comment on the zeta-potential values so close to zero at pH6-6.5 (Fig 4A in green and blue); one cannot attribute positive charges to such liposomes; they are basically neutral due to the presence of cationic and anionic lipids in the same bilayer.

We have added the following sentence in Section 2.5 to capture this feature of the near-zero z-potential: “It is worth noting that ICL aggregated with the cell membrane-mimetic liposomes even at near-zero z-potentials (DHMI and DHDMI at pH 6.5), indicating that acquisition of excessive positive charge is not necessary for ICL to adsorb onto the model liposomes. Instead, the aggregation may be attributed to the loss of negative charges on ICL surface when the positively charged imidazole lipids cluster with negatively charged DPPE-PEG.”

Reviewer 3 Report

This is one of the most refined papers I saw recently in nanomedicine.

I will reject its publication for only one reason: the authors failed to highlight their work's significance and novelty. Imidazole pH-sensitive liposomes are not a new technology, and the authors should emphasize the importance of their study. In addition, more references regarding the current development of IMZ-liposomes should be included in the discussion. Intro also is a little bit too long, sometimes seeming more like a review. Please state clearly scientific questions and the goal of the project. 

I liked the parallel investigation between particles with and without cholesterol, and I believe this is one (not the only) major novelty of this work in the field.

I hope to see this manuscript soon improved, and I will be happy to recommend it for publication.

Author Response

Reviewer 3:

We thank the reviewer for contributing precious time and for providing constructive feedback. The following are our point-by-point response to the reviewer’s comments.

“I will reject its publication for only one reason: the authors failed to highlight their work's significance and novelty. Imidazole pH-sensitive liposomes are not a new technology, and the authors should emphasize the importance of their study.”

We have made extensive modifications to highlight the significance and novelty of this week. The following are examples of such modifications.

To explain the main novelty of this work, we have added the following text in the abstract: “The imidazole lipids were designed to protonate and cluster with negatively charged PE-PEG when pH drops from 7.4 to 6.0, thereby triggering ICL in acidic tumor interstitium.”

In the discussion section, we have added the following texts to explain the novel structure feature of the imidazole lipids under this study that enables their efficient clustering with DPPE-PEG: “In this study on ICL, the three imidazole-based lipids triggered PEGylated liposomes by efficiently clustering with phospholipid-PEG conjugates. Such a feature differentiates them from the imidazole lipid reported by Ju et al [24] and represents a novel approach to construct stealth liposomes with pH-sensitivity. The clustering action is most probably achieved by the three lipids’ unique structure, in which the imidazole headgroup is linked to the lipid tail at the C2 position through a carbon-sulfur bond so that both nitrogen atoms of the imidazole group can serve as H-bond donors upon protonation at acidic pH (Figure 2). The protonated imidazole groups can then each bind with negatively charged phosphate groups from two different DPPE-PEG molecules, which in turn crosslink DPPE-PEG molecules into clusters on the ICL surface. Because PEGylation serves as a key method to construct long circulating liposomes for anticancer drug delivery by the EPR effect, the imidazole-based lipids under this study have the potential for wide applications in vivo.”

To highlight the significantly improved anticancer activities of ICL compared to the well-established PEGylated liposomal doxorubicin in current clinical use, we have added in the abstract, “ICL demonstrated superior activities against monolayer cells and several 3D MCS than the analogous PEGylated, pH-insensitive liposomes containing DOX, which serves as a control and clinical benchmark.” For the same purpose, we have also included the following text in the modified conclusion: “ICL demonstrated substantially higher anticancer activities than the analogous PEGylated, pH-insensitive liposomes containing doxorubicin, which is a common type of nano- formulations in clinical use. While the anticancer activities of ICL against monolayer Hela cells are correlated with higher pKa of the imidazole lipid, the anti-cancer activities against 3D multicellular spheroids are the highest in ICL that consisted of the imidazole lipid DHMI, which possesses the medium pKa and enhances the liposomal drug release at pH 6.0. Our studies on ICL suggest that nano- drug de-livery systems that balance the needs of intratumoral penetration, adsorption to cancer cells, and enhanced drug release would yield optimal anticancer activities.”

“In addition, more references regarding the current development of IMZ-liposomes should be included in the discussion.”

As mentioned in our response to the previous comment, we have added the following texts into the discussion section to explain the key novel structure feature of the imidazole lipids under this study that enables their efficient clustering with PE-PEG: “In this study on ICL, the three imidazole-based lipids triggered PEGylated liposomes by efficiently clustering with phospholipid-PEG conjugates. Such a feature differentiate them from the imidazole lipid reported by Ju et al [24] and represents a novel approach to construct stealth liposomes with pH-sensitivity. The clustering action is most probably achieved by the three lipids’ unique structure, in which the imidazole headgroup is linked to the lipid tail at the C2 position through a carbon-sulfur bond so that both nitrogen atoms of the imidazole group can serve as H-bond donors upon protonation at acidic pH (Figure 2). The protonated imidazole groups can then each bind with negatively charged phosphate groups from two different DPPE-PEG molecules, which in turn crosslink DPPE-PEG molecules into clusters on the ICL surface. Because PEGylation serves as a key method to construct long circulating liposomes for anticancer drug delivery by the EPR effect, the imidazole-based lipids under this study have the potential for wide applications in vivo.”

“Intro also is a little bit too long, sometimes seeming more like a review. Please state clearly scientific questions and the goal of the project.”

We have removed some of the nonessential background information from the introduction section.

We have added the following text to clarify the goal of the project: “Herein we report a novel type of imidazole lipids and their pH-sensitive liposomes. Our goal is to develop imidazole lipids that trigger the liposomes in cooperation with phosphatidylethanolamine-polyethylene glycol conjugates (PE-PEG), which is a key component to stabilize liposomes in blood circulation for anticancer drug delivery.”

We have kept the contents regarding targeting by EPR, challenge of drug penetration in solid tumors, PEG, long circulation liposome, and pH-sensitivity in the introduction, all serving as the rationale for designing such a PEGylated, pH-sensitive liposome.

“I liked the parallel investigation between particles with and without cholesterol, and I believe this is one (not the only) major novelty of this work in the field.”

We have added a couple of sentences to summarize the effects of cholesterol on ICL in the conclusion and the abstract sections. We believe that this knowledge will help the further development of ICL for anticancer applications.

“I hope to see this manuscript soon improved, and I will be happy to recommend it for publication.”

We thank the reviewer for the encouragement and hope that we have addressed all the reviewer’s concerns.

Round 2

Reviewer 3 Report

NA